# Racial and ethnic disparities in "stop-and-frisk" experience among young sexual minority men in New York City

Maria R. Khan[1,2]*, Farzana Kapadia[1,3], Amanda Geller[4], Medha Mazumdar[1], Joy D. Scheidell[1,2], Kristen D. Krause[5], Richard J. Martino[5], Charles M. Cleland[1,2], Typhanye V. Dyer[6], Danielle C. Ompad[2,3], Perry N. Halkitis[5,7]

1 Department of Population Health, New York University Grossman School of Medicine, New York, New York, United States of America, 2 Center for Drug Use and HIV Research, New York University, New York, New York, United States of America, 3 Department of Epidemiology, New York University, New York, New York, United States of America, 4 Department of Criminology, Law and Society, University of California - Irvine, Irvine, California, United States of America, 5 Center for Health, Identity, Behavior & Prevention Studies, Rutgers University, Piscataway, New Jersey, United States of America, 6 Department of Epidemiology and Biostatistics, University of Maryland School of Public Health, College Park, Maryland, United States of America, 7 Departments of Biostatistics and Epidemiology and Urban-Global Health, Rutgers University, Piscataway, New Jersey, United States of America

* maria.khan@nyulangone.org

**Data Availability Statement:** To protect the confidentiality of P18 cohort participants, we do not make data freely available. Data are housed at the Rutgers School of Public Health's Center for

## Abstract

Although racial/ethnic disparities in police contact are well documented, less is known about other dimensions of inequity in policing. Sexual minority groups may face disproportionate police contact. We used data from the P18 Cohort Study (Version 2), a study conducted to measure determinants of inequity in STI/HIV risk among young sexual minority men (YSMM) in New York City, to measure across-time trends, racial/ethnic disparities, and correlates of self-reported stop-and-frisk experience over the cohort follow-up (2014–2019). Over the study period, 43% reported stop-and-frisk with higher levels reported among Black (47%) and Hispanic/Latinx (45%) than White (38%) participants. Stop-and-frisk levels declined over follow-up for each racial/ethnic group. The per capita rates among P18 participants calculated based on self-reported stop-and-frisk were much higher than rates calculated based on New York City Police Department official counts. We stratified respondents' ZIP codes of residence into tertiles of per capita stop rates and observed pronounced disparities in Black versus White stop-and-frisk rates, particularly in neighborhoods with low or moderate levels of stop-and-frisk activity. YSMM facing the greatest economic vulnerability and mental disorder symptoms were most likely to report stop-and-frisk. Among White respondents levels of past year stop-and-frisk were markedly higher among those who reported past 30 day marijuana use (41%) versus those reporting no use (17%) while among Black and Hispanic/Latinx respondents stop-and-frisk levels were comparable among those reporting marijuana use (38%) versus those reporting no use (31%). These findings suggest inequity in policing is observed not only among racial/ethnic but also sexual minority groups and that racial/ethnic YSMM, who are at the intersection of multiple minority statuses, face disproportionate risk. Because the most socially vulnerable experience disproportionate stop-and-frisk risk, we need to reach YSMM with community resources to

Health, Identity, Behavior and Prevention Studies (CHIBPS), where P18 cohort PI (Halkitis) is housed. Data can be made available upon request through Anita Karr available at (ak1768@njms. rutgers.edu).

**Funding:** This research was funded by grants from the National Institute on Drug Abuse (R01DA044037, PI: MR Khan; R01DA0225537, PI: PN Halkitis; R01DA025537, PI: PN Halkitis). MRK, JDS, CMC, and DCO were supported by the New York University Center for Drug Use and HIV Research (P30 DA011041). MRK and JDS additionally were supported by the New York University-City University of New York (NYU-CUNY) Prevention Research Center (U48 DP005008). TVD was supported by the University of Maryland Prevention Research Center (U48 DP006382). The funders had no role in study design, data collection and analysis, decision to publish or preparation of the manuscript.

**Competing interests:** The authors have declared that no competing interests exist.

promote health and wellbeing as an alternative to targeting this group with stressful and stigmatizing police exposure.

## Introduction

Police-community relations in New York City have long been a topic of contentious debate (see Fagan et al, 2010 for a review), with investigative stops, alternatively known as "stop-and-frisk," "*Terry* stops," or "stop question and frisk" activity, a practice of particular prominence in public discourse and litigation since the 1990s [1–4]. Implemented by the New York City Police Department (NYPD) as a method of *Broken Windows* policing [5], stop-and-frisk was promoted as a tactic for enforcement of laws aimed against low-level social disorder with a parallel focus on the proactive detection and seizure of illegal guns [6]. However, despite officers recording more than 4 million stops over a ten-year period from 2004–2013 [7], the vast majority of these stops did not lead to arrest or the detection of illegal activity. Stop-and-frisk encounters are a source of stress and disempowerment [8–10] and carry the possibility of escalating to violence, raising the risk of physical injury and mortality [9, 10]. A single encounter with the police may be an acute stressor; moreover, individuals who are stopped repeatedly or live in high-stop neighborhoods, may experience this police presence as a chronic stressor.

The NYPD's stop-and-frisk program was found to violate 4$^{th}$ Amendment protections against unlawful searches and seizures, and these practices were further found to violate 14$^{th}$ Amendment protections [2, 11]; nearly 90% of individuals who were stopped were Black and Hispanic/Latinx, a disparity that exceeded what might be predicted either by race differences in either population or neighborhood crime patterns [12]. Black men and women face unacceptable brutality and death at the hands of law enforcement [13]. In response to legal action taken against the NYPD stop-and-frisk program, reported stop activity has declined by more than 90% since its 2011 peak [7]. The decline reflects both a change in policing practices and recording practices [14]. However, the independent police monitor overseeing police stops has cited underreporting of stops as a serious problem [14], and it is widely recognized that New Yorkers face far more invasive policing than official statistics indicate [14]. The magnitude of the underreporting is unclear given a dearth of research on the magnitude of stop-and-frisk in the city.

While racial disparities in policing are well documented, we know little about other dimensions of inequity in policing. There is evidence that sexual minority groups face disproportionate risk of policing; in a 2013 study conducted among approximately 2,000 LGBTQ persons, almost three-quarters had had contact with the police in the past five years [15]. Within that sample, approximately 20% reported that the police had a hostile attitude, with about 10% being physically searched and 14% being verbally assaulted; those who identified as a person of color were about two times more likely to report these experiences [15]. There has been scant research on the burden of police contact faced by sexual minority men. Given evidence of the disproportionate burden of police exposure among sexual minority groups, a study measuring levels of stop-and-frisk experience that also assesses risk of police exposure among those at the intersection of both sexual and racial/ethnic minority status is warranted.

The purpose of the current study was to describe levels of and factors associated with self-reported stop-and-frisk experience in young sexual minority men (YSMM) in New York City. We used data collected during the P18 Cohort Study (Version 2), a study conducted to measure determinants of inequity in STI/HIV risk in a racially and ethnically diverse cohort of

young sexual minority men (YSMM) in New York City. To address the gap in the literature on burden of stop-and-frisk in YSMM, we aimed to describe levels and trends of stop-and-frisk experience over the cohort follow-up period (2014 to 2019). We hypothesized we would observe a substantial proportion of YSMM have a history of stop-and-frisk experience, with a disproportionate burden in racial/ethnic YSMM. We also hypothesized we would observe decreases in stop-and-frisk over time consistent with the decreases in reported stop-and-frisk activity by NYPD yet that across-time racial/ethnic disparities in stop-and-frisk would persist. To address concerns that official counts of stop-and-frisk in NYC based on NYPD records do not reflect the experience of stop-and-frisk burden in the population we aimed to compare rates of self-reported stop-and-frisk experience in the P18 cohort with stop-and-frisk rates among young men based on NYPD official reports, hypothesizing rates of police contact would be higher in the P18 sample. To better understand racial/ethnic disparities in stop-and-frisk of YSMM and where in NYC racial/ethnic and sexual minority men faced the greatest inequity in police exposure, we sought to describe the geographic distribution of police exposure and disparities therein. We hypothesized that respondents of all racial/ethnic backgrounds living in higher-stop areas would have had high risk of prior contact with the police [12], while racial/ethnic disparities in stop-and-frisk would be pronounced in neighborhoods where stop-and-frisk activity was less prevalent overall. Finally, to best understand the social and health needs of YSMM who come into contact with the police, we also assessed socio-demographic, psychosocial vulnerability, and behavioral correlates of stop-and-frisk. We hypothesized YSMM with the greatest social vulnerability (poverty, mental health symptoms) would have had greatest risk of contact with the police. We also hypothesized drug use would be a strong correlate of stop-and-frisk among White YSMM but not among minority YSMM given disproportionate stop-and-frisk risk in minority YSMM including among those who do not endorse drug use.

## Materials and methods

### Data

**P18 cohort data.** We used data from the second wave of the Project 18 Cohort study [16, 17], a cohort (N = 665) established in the New York City metropolitan area to evaluate determinants of STI/HIV, drug use, and mental health burden and the confluence of these health conditions as per a model of syndemics in YSMM [18, 19]. Approximately half of the participants who took part in the first wave of the P18 Cohort Study (2009–2014) were included in the second wave (n = 274). We then recruited additional participants (n = 391) matched to the age of those from wave 1 who met the following criteria: were age 22–26 years, were assigned male sex at birth, reported sex with a man in the prior six months, reported negative or unknown HIV status, resided in the New York City metropolitan area, and provided written informed consent. Prescreening to determine eligibility was performed either in person or over the telephone. At baseline and at the six-, 12-, 18-, 24-, and 36-month follow-up visits, participants completed an audio computer-assisted self-interview (ACASI) survey that assessed socio-demographic factors, stop-and-frisk experiences, psychosocial vulnerability factors, symptoms of depression and anxiety, and behavioral factors. The interquartile ranges for years when surveys were administered were: baseline (2014–2015), 12-month (2015–2016), 24-month (2016–2017), and 36-month (2017–2018) follow-up. The final 36-month survey was administered in 2019. Institutional review boards at New York University (School of Medicine and College of Global Public Health) approved this secondary data analysis study using P18 data. Participation of investigators from other institutions was considered non-human subjects research activity.

**Administrative data.** *Comparative Analyses.* We compared stop-and-frisk in YSMM versus the General Population. We used 2014–2018 data from the New York City Police Department (NYPD) Stop, Question, and Frisk database [7] with population data from the 2014–2018 American Community Survey to calculate per capita stop-and-frisk rates in the general-population of males in New York City [20]. In New York City, the expectation by the NYPD is that each time law enforcement stops an individual, the officer is to complete a hand-written record of the stop. Each stop is to be manually entered into an NYPD database which is released annually to the public. The American Community Survey, implemented by the US Census Bureau, provides annual population estimates for US cities and towns to enable communities to track changes in population distributions.

*Contextual Analyses.* We evaluated racial/ethnic disparities in stop-and-frisk in YSMM, by neighborhood-level stop activity. Spatial data indicating the boundaries of each of the 77 NYPD police precincts were imported from the New York City Department of Planning [21]. P18 participants were assigned to their precinct of residence based on the centroid of their self-reported ZIP code.

## Measures

**P18 cohort: Stop-and-frisk experience.** At each study visit participants were asked "How many times have you been stopped, questioned, and/or frisked in the past year?" Because the question assessed past year experience, we analyzed the data collected annually, at baseline and the 12-, 24-, and 36-month follow-up visits. Using this questionnaire item we examined a range of stop-and-frisk indicators including a dichotomous indicator of whether a participant reported any past year stop-and-frisk experience, and the number of incidents each participant reported experiencing. The latter measure was coded into a dichotomous indicator of repeat (two or greater) stop-and-frisk experience, and was used to compute a group-level stop-and-frisk rate, calculated as the total number of stop-and-frisk events reported by Black, Hispanic, and White participants, respectively, divided by the number of P18 study participants in each group. The reported number of incidents was also used to compute a race by neighborhood stratum rate, based on the number of incidents reported, and number of Black, Hispanic and White participants living in low, medium, and high-stop neighborhoods.

**P18 cohort: Socio-demographic, psychosocial, and behavioral factors.** We examined levels of stop-and-frisk experience by race/ethnicity, by ZIP code, and across study year. Because Black race is strongly associated with disproportionate policing, those who identified as Black including Black YSMM who also endorsed Hispanic/Latinx ethnicity were categorized as Black.

We measured correlations between baseline report of any past year stop-and-frisk and repeat stop-and-frisk and the following factors also measured at baseline: gender (male, female, transgender, gender queer); age; New York City borough of residence and borough reported as the place respondent is most likely to hang out; sexual identity (heterosexual, gay/lesbian, bisexual); education; employment; income; unstable housing; psychosocial vulnerability factors including internalized homophobia (Range: 4–20; dichotomized at ≥44) [22], depression measured using the Beck Depression Inventory (Range: 0–55, categorized as minimal (<14), mild (14–20), moderate (21–28), severe (≥29)) [23], Beck's Anxiety Inventory (Range: 0–63; categorized as minimal (<10), mild (10–18), moderate (19–29), severe (≥30)) [24]; post-traumatic stress disorder measured using the 17-item PCL-C checklist (Range: 17–84; dichotomized at ≥44) [25], self-report of neglect by the time the participant started 6th grade; any physical and/or sexual abuse in childhood; self-reported past 30 day substance use including binge drinking, use of marijuana, and any "hard" drugs (i.e., cocaine, crack,

methamphetamine, opioid, stimulants, Rohypnol, or GHB) measured using the timeline fol-low-back method.

**New York City young male stop-and-frisk rate.** Trends in race-specific YSMM rates of stop-and-frisk per 1,000 population over P18 cohort follow-up (at baseline and the 12-, 24-, and 36-month follow-ups, which took place from 2014–2019 with the majority of 36 month follow-ups occurring from 2015–2018) were compared to trends in race-specific general-pop-ulation young male stop-and-frisk rates from 2014–2018. Annual age, sex, and race-specific, population-level rates of stop-and-frisk per 1,000 population, were calculated as the annual age, sex, and race-specific number of stop-and-frisk events recorded by the NYPD divided by the age, sex, and race-specific estimate of the New York City population recorded in the Amer-ican Community Survey, multiplied by 1,000. Reflecting the P18 focus on young men, rates were computed for the male population aged 20–29 years.

**Contextual factor: Neighborhood-level stop-and-frisk activity.** We examined whether racial/ethnic disparities in stop-and-frisk levels among P18 participants varied depending on whether the respondent lived in a neighborhood with a low, moderate, or high rate of stop activity. To calculate neighborhood-level stop activity, P18 participant ZIP code was trans-lated to latitude and longitude, which was then used to identify the New York City Police Department (NYPD) precinct in which the respondent lived. NYPD precinct-level stop-and-frisk rates were calculated as the 2014 NYPD recorded number of stop-and-frisk events among 20–29 year old men, divided by the precinct population of male residents aged 20–29 years old. The precinct population was estimated using population of the census tract, deter-mined by the 2014 American Community Survey, that had the greatest spatial overlap with the precinct; population estimates for each census tract of New York City were calculated after restricting the sample to males aged 20–29 years. Each of the 77 precincts were catego-rized into three group groups based on the precinct-level stop-and-frisk tertile, which served as an indicator of neighborhoods with low, moderate, and high levels of stop activity. Within each tertile, race-specific P18 stop-and-frisk rates were calculated as the race-specific number of stop-and-frisk events reported by P18 participants divided by the race-specific P18 study population.

## Statistical analysis

We summarized the prevalence of self-reported police contact and reported stop-and-frisk rate by race/ethnicity at baseline and the 12, 24, and 36-month follow-up visits. Using Poisson regression, we estimated unadjusted rate ratios (RRs) and 95% confidence intervals for associa-tion between race (Hispanic/Latinx and Black vs White, the referent) and contact rate. To compare levels of stop-and-frisk among YSMM to other men of comparable age in New York City, we plotted trends in stop-and-frisk rates in the P18 sample versus the New York City young male stop-and-frisk rate. Using participant ZIP code, we mapped the spatial distribu-tion of P18 participants by past year stop-and-frisk frequency; maps were created separately by race/ethnicity to allow for comparison of the spatial differences in stop-and-frisk experience across groups.

We further investigated racial/ethnic disparities in stop-and-frisk levels by measuring whether racial/ethnic disparities in stop-and-frisk levels among P18 participants varied depending on whether the respondent lived in a neighborhood with a low, moderate, or high rate of stop activity. Using Poisson regression, we estimated unadjusted RRs and 95% confi-dence intervals for associations between race (Hispanic/Latinx and Black vs White, the refer-ent) within each tertile. To understand the factors that might place participants at risk for police contact, we used logistic regression to estimate adjusted odds ratios (ORs) and 95%

confidence intervals for associations between participant characteristics and past year history of any prior and repeat stop-and-frisk experience.

## Results

### Levels of stop-and-frisk by race/ethnicity among YSMM

Over the study period, 43% of P18 participants reported a history of stop-and-frisk experience in the prior year, with higher levels among Black (47%) and Hispanic/Latinx (45%) than White (38%) participants.

As shown in Fig 1a, at baseline, the prevalence of any self-reported stop-and-frisk experience for the 12-months preceding study interview was higher among Black (35%) and Hispanc/Latinx (34%) YSMM compared to their White (31%) counterparts; in the sample overall 33% experienced stop-and-frisk in the past year. While there were declines in the prevalence of stop-and-frisk encounters for all groups between the baseline and 12-month follow-up visit, the racial disparities in stop-and-frisk evident at the baseline visit remained at the 12-month follow-up visit and persisted over the remaining study period despite declines in all three groups. Specifically, for White and Hispanic/Latinx YSMM, there was almost a 65% (31% to 11% and 34% to 12%, respectively) decrease in stop-and-frisk over the study period. However, among Black YSMM, the decline in stop-and-frisk encounters was only 54% (35% to 16%). Most importantly, at every visit during the study period, prevalence of any stop-and-frisk reports were higher among Black YSMM than among Hispanic/Latinx and White YSMM, with Black-White differences consistently statistically significant. There was substantial heterogeneity in the levels of past year stop-and-frisk among those who endorsed stop-and-frisk over time. The maximum number of stops per person was markedly higher among Black and Hispanic/Latinx vs White respondents (Black: 20, Hispanic: 23, White: 12 stops) and remained disparate over follow-up through the 36-month follow-up visit (Black: 66, Hispanic: 5, White: 2 stops). Of note, the Black participant with 66 stops was an outlier who also reported high scores on measures of personal stigma, public stigma, internalized homophobia, and PTSD symptoms. We discuss the sensitivity of our findings to this participant's inclusion below.

Next, we examined per-participant stop rates (i.e., the reported number of encounters per participant) across racial/ethnic groups. As shown in Fig 1b, participant experiences of stop-and-frisk decreased between the baseline visit and the 36-month follow-up visit. However, these decreases in rates of stop-and-frisk were not consistent across race/ethnicity. Specifically, among White YSMM, the rate of stop-and-frisk decreased by 83% from 0.6 to 0.1 from baseline to the end of the study period. A similar trend was observed among Hispanic/Latinx YSMM with stop-and-frisk rates decreasing by 80% from 1 to 0.2 over the study period. However, among Black YSMM, rates of stop-and-frisk encounters did not decline consistently over the study period. Black participants experienced a comparable decline in per-participant stop rate (75% decline; from 1.2 to 0.3 encounters per person) through the 24-month follow-up, but a considerable uptick at the 36-month interview (66% increase; from 0.3 to 0.9 encounters per person). This increase was driven in part by a small number of respondents reporting very high stop rates (e.g., the aforementioned participant reporting 66 times stopped in the prior year at the 36-month follow-up) but suggests that the decline was not sustained for Black participants to the extent it was for those who were White or Hispanic/Latinx. Black participants had twice the stop-and-frisk rate as White participants at the baseline visit (RR: 2.09, 95% CI: 1.68, 2.63) and 12-month visit (RR: 2.13, 95% CI: 1.46, 3.18). By the 36-month follow-up the disparity grew to five times the rate (RR: 5.89, 95% CI: 3.63, 10.28); yet when we removed from the analysis one Black participant who reported 66 stop-and-frisk experiences the stark disparity remained (RR: 3.15, 95%: 1.89, 5.59).

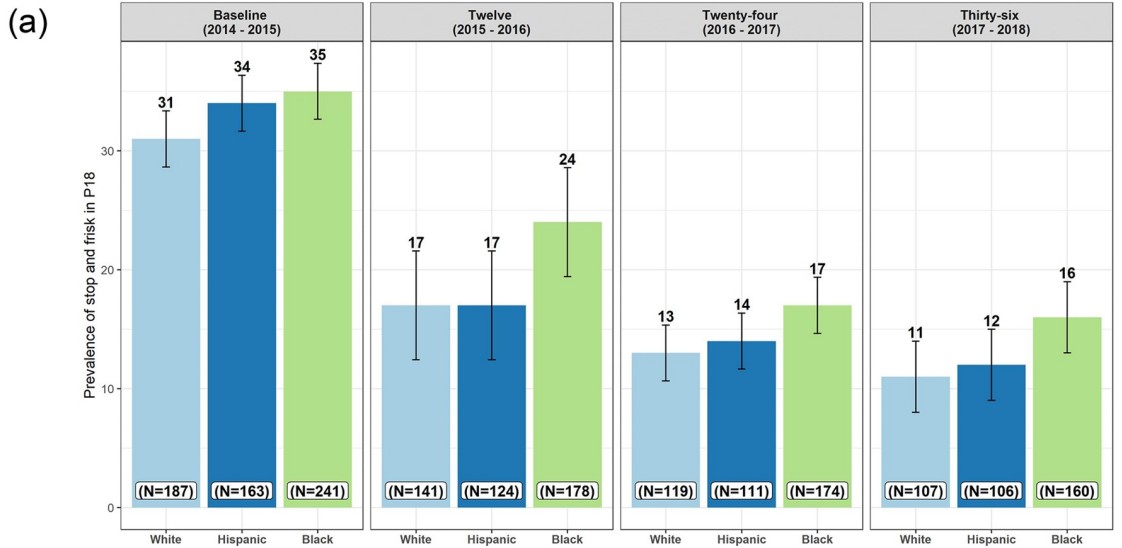

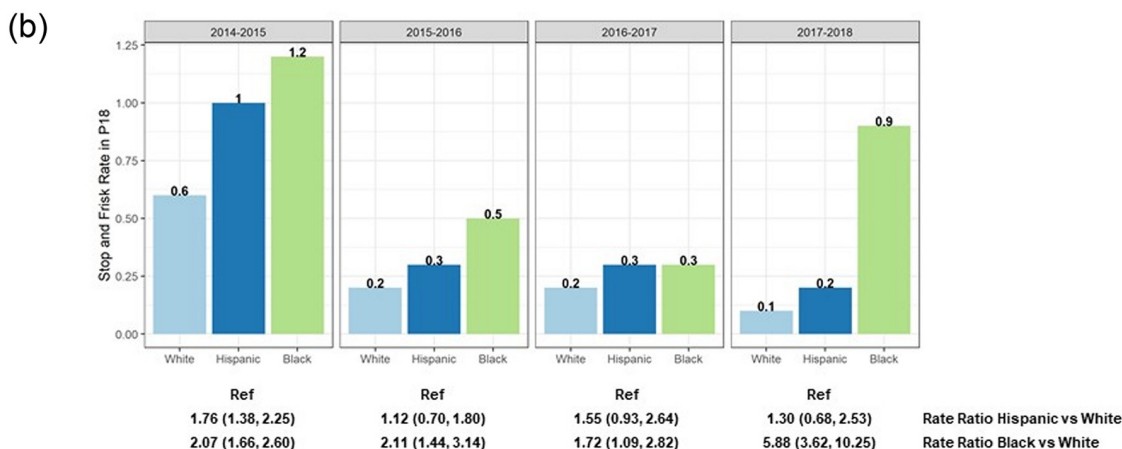

**Fig 1. a.** Race differences in past year stop-and-frisk prevalence of past year stop-and-frisk* among P18 participants over cohort follow-up (2014–2019; N = 591 at baseline). *P18 questionnaire item: "How many times have you been stopped, questioned, and/or frisked in the past year?" We coded a dichotomous indicator of any prior stop-and-frisk in the past year. In the sample overall, 32.3% reported being stopped-and-frisk at baseline and 21.5% reported being stopped-and-frisked at least once over the course of the follow-up. **b.** Race differences in past year stop-and-frisk rate* among P18 participants over cohort follow-up (above bar chart) and unadjusted rate ratios (RRs) and 95% confidence intervals (CIs) for associations between race and stop-and-frisk rate (below table) (2014–2019; N = 591 at baseline). *P18 questionnaire item: "How many times have you been stopped, questioned, and/or frisked in the past year?" P18 rates were calculated as the race-specific number of stop-and-frisk events reported by P18 participants/race-specific P18 study population times 1,000. Note: Rate among Black P18 participants at the 36-month visit was driven by a high stop rate among one participant (66 stops in the past year); when this participant was excluded from the analysis the stop rate was reduced to 0.5 from 0.9 among the remaining participants.

### Levels of self-reported stop-and-frisk among YSMM versus official reports in the general population

In Fig 2 we compare rates of per capita stop-and-frisk experience among study participants to that of young men in New York City at large. Notably, the overall decline in stop-and-frisk in our study sample parallel the population-level decline in stop-and-frisk among males ages 20–

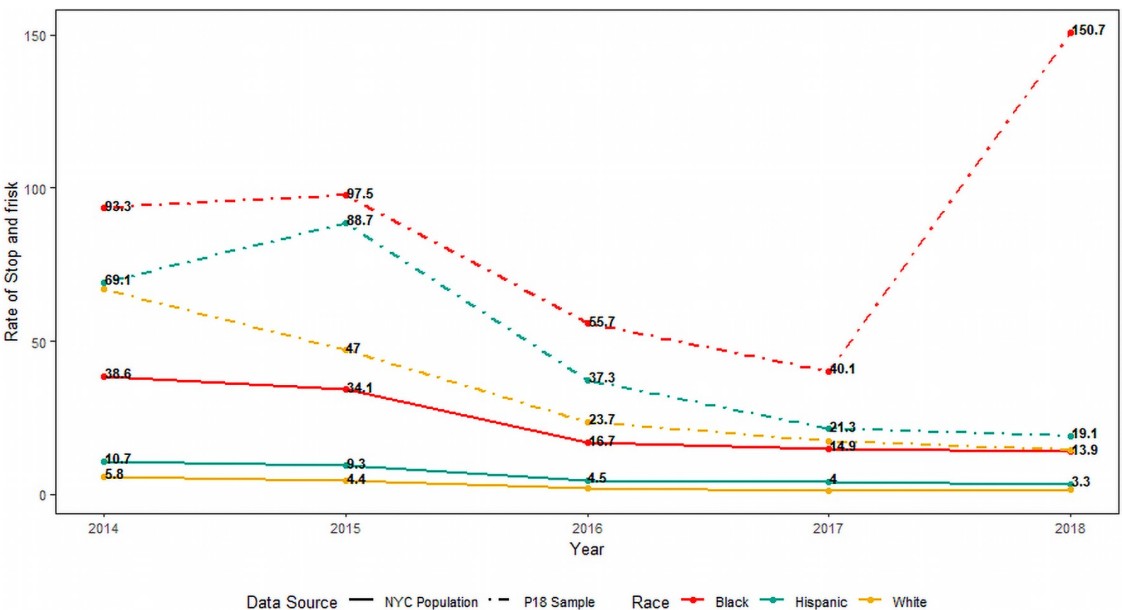

**Fig 2. Stop-and-frisk rate per 1,000 population*** **reported by P18 participants vs recorded by the New York City Police Department (NYPD), by race.** *P18 rates were calculated as the race-specific number of stop-and-frisk events reported by P18 participants/race-specific P18 study population. These were compared to rates in the general population, calculated as the race-specific number of stop-and-frisk events recorded by the New York City Police Department/race-specific American Community Survey New York City population estimates. Note: Rate among Black P18 participants at the 36-month visit was driven by a high stop rate among one participant (66 stops in the past year); when this participant was excluded from the analysis the stop rate was reduced to 58.6 per 1,000.

29 years old during this same period. Racial/ethnic disparities are similarly comparable to those at the population level, though the magnitudes of these disparities are less extreme. However, the per capita rate of stops reported by P18 participants of all races is significantly higher than that recorded citywide, in some cases by orders of magnitude.

## Spatial distribution of stop-and-frisk in New York City among YSMM, by race/ethnicity

Fig 3 provides the geographic distribution of P18 respondents based on their reported ZIP codes of residence and indicates that at baseline, P18 participants residing throughout all five New York City boroughs including Manhattan and the four outer boroughs (Brooklyn, Queens, the Bronx, and Staten Island) reported stop-and-frisk. Observed racial disparities in self-reported stop-and-frisk experiences may have been driven in neighborhood exposure. White P18 respondents were more likely to live in Manhattan than their Black and Hispanic/Latinx counterparts, who were more likely to live in high-stop areas of the Brooklyn and the Bronx, which are neighborhoods that have historically experienced high rates of aggressive policing [12]. Stop experience among Black and Hispanic/Latinx participants was not limited to those living in high-stop neighborhoods; those living outside of New York City also reported having been stopped.

## Racial/ethnic disparities in stop-and-frisk, by neighborhood-level stop activity

Given the likely role of neighborhood policing in participants' exposure to police encounters, we examined racial/ethnic differences in their reported rates of stop-and-frisk (i.e., reported

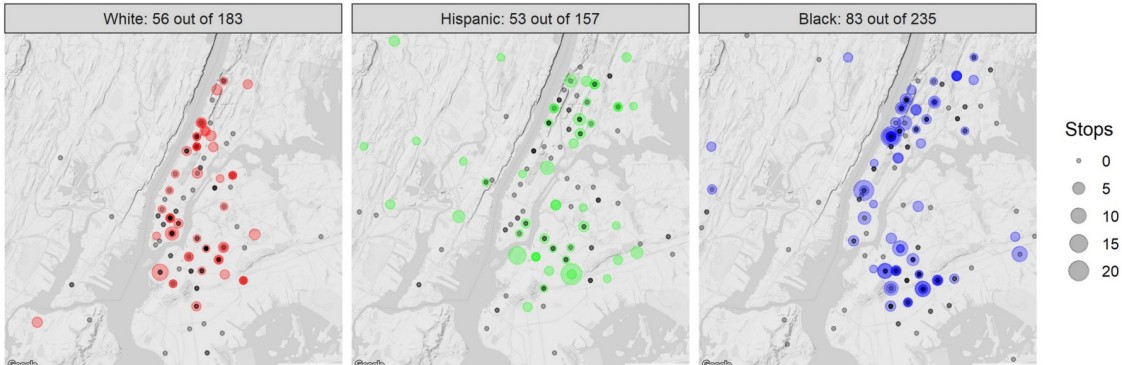

**Fig 3. Geographic distribution of past year stop-and-frisk among P18 participants at cohort baseline, by race: P18 participants are mapped by participant ZIP code of residence.** Those with a history of stop-and-frisk are indicated by red circles (White race), green circles (Hispanic/Latinx race/ethnicity), blue circles (Black race) with increasing circle size indicating increasing number of stop-and-frisk events, while those with no stop-and-frisk history are indicated by Black circles.

baseline stops per participant) among study participant by tertiles of neighborhood-level rates (Fig 4). Within the lowest stop New York City neighborhoods (i.e., the lowest tertile), White (0.45) and Hispanic/Latinx (0.42) YSMM had substantially lower stop-and-frisk rates than Black YSMM (1.2). In neighborhoods with the second-highest tertile of neighborhood-level stop-and-frisk rates, White YSMM had the lowest rate of stop-and-frisk encounters (0.51) compared to their Hispanic/Latinx (0.78) and Black (1.33). However, in neighborhoods

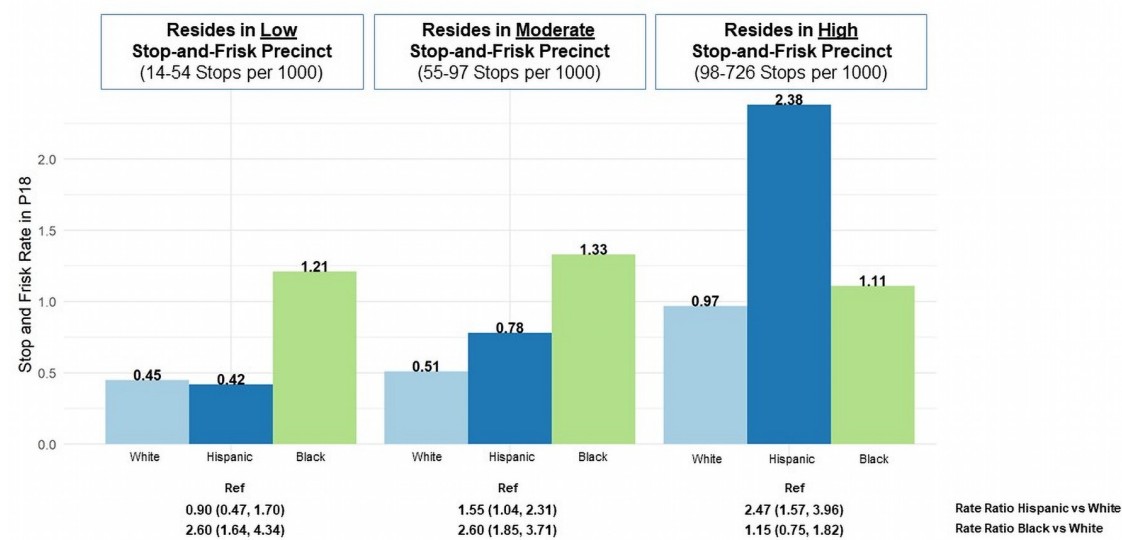

**Fig 4. Race differences in past year stop-and-frisk rate among P18 participants at cohort baseline in 2014 by neighborhood-level stop-and-frisk rate (counts/1000)\*.** \*P18 participants were categorized according to their neighborhood-level stop-and-frisk rate, which were calculated as followed. Participant's ZIP code was translated to latitude and longitude, which was then used to identify the New York City Police Department (NYPD) precinct in which the respondent lived (total of 77 possible precincts). NYPD precinct-level stop-and-frisk rates were calculated as the 2014 NYPD recorded number of stop-and-frisk events divided by the precinct population of male residents aged 20–29 years old; restricted given the younger age range of the P18 sample. The population estimates for New York City were calculated from 2014 estimates from the 2010–2014 5-year American Community Survey; population estimates for each census tract of New York City were calculated after restricting the sample to males aged 20–29 years. Each of the 77 precincts were categorized into group groups based on the precinct-level stop-and-frisk tertile. Within each tertile indicating neighborhood-level stop-and-frisk, stop-and-frisk rates were calculated at the race-specific number of stop-and-frisk events reported by P18 participants/race-specific P18 study population.

comprising the highest tertile of neighborhood-level stop-and-frisk rates, rates of stop-and-frisk encounters among White YSMM (0.92) were only slightly lower than that among Black YSMM (1.11). Moreover, rates of stop-and-frisk encounters among Hispanic/Latinx (2.38) YSMM were more than two times higher than among their White and Black YSMM. In neighborhoods with low and moderate stop-and-frisk rates, Black participants had over 2.5 times the stop-and-frisk rates (lowest tertile neighborhood stop-and-frisk: RR 2.66, 95% CI: 1.68, 4.44; middle tertile neighborhood stop-and-frisk: RR: 2.63, 95% CI: 1.87, 3.75) while in the highest stop-and-frisk tertile Black and White P18 participants had comparable rates (RR: 1.15, 95% CI: 0.75, 1.82).

### Individual-level factors associated with stop-and-frisk

Finally, in analyses assessing associations between stop-and-frisk and socio-demographic, psychosocial and substance use factors (Table 1), the odds of multiple stop-and-frisk encounters among Black and Hispanic/Latinx YSMM were almost 2 times higher than for White YSMM (OR = 2.35, 95% CI 1.39, 4.08 and OR = 1.82, 95% CI 1.02, 3.32, respectively). Next, while 2% of participants reported they did not identify with a binary gender, the odds of any stop-and-frisk encounter as well as multiple encounters were 2 to almost 5 times higher in this group (OR = 3.51, 95% CI 1.15, 11.7 and OR = 5.78, 95% CI 1.88, 18.28, respectively). Among YSMM who reported being more likely to socialize or hang out in the Bronx, the odds of both any and multiple stop-and-frisk encounters where higher (OR = 2.90, 95% CI 1.57, 5.43 and OR = 2.76, 95% CI 1.42, 5.25, respectively) than among YSMM reporting socializing in other New York City boroughs.

In terms of psychosocial factors, YSMM reporting higher levels of internalized homophobia had increased odds of reporting both any and multiple stop-and-frisk encounters (OR = 1.58, 95% CI 1.05, 2.37 and OR = 1.64, 95% CI 1.01, 2.63, respectively). With respect to mental health burdens, YSMM who reported any and multiple stop-and-frisk encounters had consistently higher odds of severe levels of depressive symptoms, severe levels of anxiety, and experiences of childhood victimization.

With regard to recent substance use, YSMM who reported using marijuana in the 30 days preceding study visits had higher odds of any and multiple stop-and-frisk encounters (OR = 1.97, 95% CI 1.41, 2.79 and OR = 1.79, 955 CI 1.18, 2.75). Similar relationships between any and multiple stop-and-frisk encounters were observed among individuals who reported hard drug use. Finally, the odds of multiple stop-and-frisk encounters were higher among YSMM who reported recent cocaine use (OR = 1.92, 95% CI 1.06, 3.34). While among White respondents levels of past year stop-and-frisk were markedly higher among those who reported past 30 day marijuana use (41%) than those reporting no use (17%), among Black and Hispanic/Latinx respondents, levels were comparable regardless of past 30 day marijuana use (marijuana use: 38%, no marijuana use: 31%) (race differences in associations between drug use and stop-and-frisk not presented in Table 1).

## Discussion

This study is among the first to measure the prevalence of recent and repeat stop-and-frisk experience in a population-based sample [8], and the only study to our knowledge to measure stop-and-frisk in YSMM. We observed that being stop-and-frisked by the police is a normative life event among YSMM in New York City with one in three YSMM having experienced the event at baseline, primarily administered in 2014–2015, with comparable levels of the event reported among Black, non-Black Hispanic/Latinx, and White participants. While levels decreased over time, consistent with documented city-wide trends in stop-and-frisk [7], at the

**Table 1. Odds ratios (ORs) and 95% confidence intervals (CIs) for associations between participant socio-demographic, psychosocial, and behavioral factors and past year stop-and-frisk at cohort baseline in 2014.**

| | Total N (%) (n = 665) | N (%) Past Year Stop-and-frisk (n = 214) | Unadjusted ORs: Association with Past Year Stop-and-frisk | N (%) Repeat (≥2) Past Year Stop-and-frisk (n = 119) | Unadjusted ORs: Association with Repeat (≥2) Past Year Stop-and-frisk |
|---|---|---|---|---|---|
| **Gender** | | | | | |
| Male | 621 (93.4%) | 194 (31.2%) | Ref | 104 (16.7%) | Ref |
| Female | 4 (0.6%) | 1 (25.0%) | 0.73 [0.04,5.74] | 0 (0%) | - - |
| Transgender female | 5 (0.8%) | 2 (40.0%) | 1.46 [0.19,8.88] | 1 (20%) | 1.24 [0.06, 8.47] |
| Genderqueer | 22 (3.3%) | 9 (40.9%) | 1.64 [0.66,3.95] | 7 (31.8%) | 2.48 [0.92, 6.11] |
| I do not identify with a gender | 13 (2.0%) | 8 (61.5%) | 3.51 [1.15, 11.7] | 7 (53.8%) | 5.78 [1.88, 18.28] |
| **Age** | | | | | |
| 21–22 | 41 (6.2%) | 13 (31.7%) | Ref | 8 (19.5%) | Ref |
| 22–23 | 320 (48.1%) | 110 (34.4%) | 1.09 [0.55, 2.26] | 59 (18.4%) | 0.90 [0.41, 2.20] |
| 23–24 | 290 (43.6%) | 87 (30.0%) | 0.90 [0.45, 1.88] | 49 (16.9%) | 0.82 [0.37, 2.01] |
| 24–26 | 13 (2.0%) | 4 (30.8%) | 0.83 [0.20, 3.03] | 3 (23.1%) | 1.09 [0.21, 4.57] |
| **Race** | | | | | |
| White | 187 (28.1%) | 58 (31.0%) | Ref | 22 (11.8%) | Ref |
| Hispanic/Latinx | 163 (24.5%) | 56 (34.4%) | 1.16 [0.74, 1.81] | 32 (19.6%) | 1.82 [1.02, 3.32] |
| Black | 241 (36.2%) | 85 (35.3%) | 1.19 [0.79, 1.80] | 58 (24.1%) | 2.35 [1.39, 4.08] |
| Other | 68 (10.2%) | 14 (20.6%) | 0.57 [0.28, 1.08] | 7 (10.3%) | 0.85 [0.32, 2.00] |
| **Borough of residence** | | | | | |
| Manhattan | 212 (31.9%) | 64 (30.2%) | Ref | 37 (17.5%) | Ref |
| Brooklyn | 203 (30.5%) | 66 (32.5%) | 1.11 [0.74,1.69] | 32 (15.8%) | 0.89 [0.53, 1.49] |
| Bronx | 93 (14.0%) | 34 (36.6%) | 1.35 [0.8, 2.25] | 21 (22.6%) | 1.39 [0.75, 2.52] |
| Queens | 61 (9.2%) | 15 (24.6%) | 0.75 [0.38, 1.41] | 7 (11.5%) | 0.61 [0.24, 1.37] |
| Staten Island | 8 (1.2%) | 1 (12.5%) | 0.33 [0.02, 1.9] | 1 (12.5%) | 0.67 [0.04, 3.94] |
| Outside **NEW YORK CITY** | 25 (3.8%) | 5 (20%) | 0.57 [0.18, 1.49] | 2 (8%) | 0.41 [0.06, 1.47] |
| Outside NYS | 59 (8.9%) | 27 (45.8%) | 1.94 [1.07, 3.5] | 18 (30.5%) | 2.06 [1.05, 3.96] |
| **Borough you like to hang out the most in** | | | | | |
| Manhattan | 424 (63.8%) | 131 (30.9%) | Ref | 74 (17.5%) | Ref |
| Brooklyn | 139 (20.9%) | 42 (30.2%) | 0.98 [0.64 1.47] | 21 (15.1%) | 0.85 [0.49, 1.41] |
| Bronx | 47 (7.1%) | 26 (55.3%) | 2.90 [1.57, 5.43] | 17 (36.2%) | 2.76 [1.42, 5.25] |
| Queens | 28 (4.2%) | 5 (17.9%) | 0.48 [0.16, 1.21] | 2 (7.1%) | 0.36 [0.06, 1.25] |
| Staten Island | 2 (0.3%) | 0 (0.0%) | 0.00 | 0 (0%) | 0 |
| **Sexual Identity** | | | | | |
| Heterosexual | 6 (0.9%) | 2 (33.3%) | Ref | 2 (33.3%) | Ref |
| Gay/Lesbian | 544 (81.8%) | 165 (30.3%) | 0.88 [0.17, 6.36] | 86 (15.8%) | 0.38 [0.07, 2.75] |

*(Continued)*

**Table 1.** (Continued)

| | Total N (%) (n = 665) | N (%) Past Year Stop-and-frisk (n = 214) | Unadjusted ORs: Association with Past Year Stop-and-frisk | N (%) Repeat (≥2) Past Year Stop-and-frisk (n = 119) | Unadjusted ORs: Association with Repeat (≥2) Past Year Stop-and-frisk |
|---|---|---|---|---|---|
| Bisexual | 103 (15.5%) | 44 (42.7%) | 1.49 [0.28, 11.11] | 28 (27.2%) | 0.75 [0.14, 5.60] |
| **Education** | | | | | |
| High school graduate or below | 318 (47.8%) | 113 (35.5%) | Ref | 73 (23%) | Ref |
| College graduate | 346 (52.0%) | 101 (29.2%) | 0.75 [0.54, 1.04] | 46 (13.3%) | 0.52 [0.34, 0.77] |
| **Employment** | | | | | |
| Unemployed/not regular job | 163 (24.5%) | 50 (30.7%) | Ref | 27 (16.6%) | Ref |
| Employed | 497 (74.7%) | 162 (32.6%) | 1.09 [0.75,1.61] | 91 (18.3%) | 1.13 [0.71, 1.83] |
| **Income** | | | | | |
| Less than $10k | 296 (44.5%) | 105 (35.5%) | Ref | 58 (19.6%) | Ref |
| More than $10k | 337 (50.7%) | 103 (30.6%) | 0.79 [0.57,1.10] | 58 (17.2%) | 0.85 [0.56, 1.27] |
| **Unstable housing** | | | | | |
| No | 580 (87.2%) | 178 (30.7%) | Ref | 96 (16.6%) | Ref |
| Yes | 60 (9.0%) | 22 (36.7%) | 1.30 [0.74, 2.24] | 15 (25%) | 1.67 [0.87, 3.05] |
| **Internalized Homophobia*** | | | | | |
| Low (<12) | 530 (79.7%) | 159 (30%) | Ref | 85 (16%) | Ref |
| High (≥12) | 121 (18.2%) | 49 (40.5%) | 1.58 [1.05,2.37] | 29 (24%) | 1.64 [1.01, 2.63] |
| **Depression (Beck Depression Inventory)** | | | | | |
| Minimal (<14) | 490 (73.7%) | 149 (30.4%) | Ref | 76 (15.5%) | Ref |
| Mild (14–20) | 82 (12.3%) | 23 (28%) | 0.89 [0.52,1.48] | 15 (18.3%) | 1.22 [0.64, 2.19] |
| Moderate (21–28) | 41 (6.2%) | 15 (36.6%) | 1.32 [0.66,2.53] | 10 (24.4%) | 1.75 [0.79, 3.61] |
| Severe (≥29) | 22 (3.3%) | 15 (68.2%) | 4.89 [2.02, 13.03] | 9 (40.9%) | 3.76 [1.50, 9.03] |
| **Anxiety (Beck's Anxiety Inventory)** | | | | | |
| Minimal (<9) | 464 (69.8%) | 143 (30.8%) | Ref | 79 (17%) | Ref |
| Mild (10–18) | 112 (16.8%) | 33 (29.5%) | 0.93 [0.59, 1.45] | 19 (17%) | 0.99 [0.56, 1.69] |
| Moderate (19–29) | 49 (7.4%) | 19 (38.8%) | 1.41 [0.76, 2.57] | 9 (18.4%) | 1.09 [0.48, 2.24] |
| Severe (≥30) | 23 (3.5%) | 13 (56.5%) | 2.90 [1.25, 6.94] | 8 (34.8%) | 2.59 [1.01, 6.17] |
| **Post-traumatic Stress Disorder (PCL-17)**** | | | | | |
| No (<44) | 543 (81.7%) | 165 (30.4%) | Ref | 91 (16.8%) | Ref |
| Yes (≥44) | 66 (9.9%) | 29 (43.9%) | 1.79 [1.06, 3.00] | 18 (27.3%) | 1.85 [1.01, 3.28] |
| **Childhood abuse (Neglect/ physical/sexual)** | | | | | |

(Continued)

Table 1. (Continued)

| | Total N (%) (n = 665) | N (%) Past Year Stop-and-frisk (n = 214) | Unadjusted ORs: Association with Past Year Stop-and-frisk | N (%) Repeat (≥2) Past Year Stop-and-frisk (n = 119) | Unadjusted ORs: Association with Repeat (≥2) Past Year Stop-and-frisk |
|---|---|---|---|---|---|
| No | 277 (41.7%) | 70 (25.3%) | Ref | 36 (13%) | Ref |
| Yes | 374 (56.2%) | 140 (37.4%) | 1.76 [1.25,2.49] | 81 (21.7%) | 1.84 [1.21, 2.85] |
| **Binge drinking in past 30 days** | | | | | |
| No | 155 (23.3%) | 44 (28.4%) | Ref | 25 (16.1%) | Ref |
| Yes | 430 (64.7%) | 148 (34.4%) | 1.32 [0.89,1.98] | 81 (18.8%) | 1.20 [0.74, 2.00] |
| **Marijuana use in past 30 days** | | | | | |
| No | 287 (43.2%) | 69 (24%) | Ref | 38 (13.2%) | Ref |
| Yes | 378 (56.8%) | 145 (38.4%) | 1.97 [1.41,2.79] | 81 (21.4%) | 1.79 [1.18, 2.75] |
| **Cocaine use in past 30 days** | | | | | |
| No | 597 (89.8%) | 185 (31%) | Ref | 100 (16.8%) | Ref |
| Yes | 68 (10.2%) | 29 (42.6%) | 1.64 [0.98,2.73] | 19 (27.9%) | 1.92 [1.06, 3.34] |
| **Hard drugs in past 30 days**\*\*\* | | | | | |
| No | 570 (85.7%) | 173 (30.4%) | Ref | 94 (16.5%) | Ref |
| Yes | 95 (14.3%) | 41 (43.2%) | 1.73 [1.11,2.69] | 25 (26.3%) | 1.80 [1.07, 2.95] |

\*IHF scale

\*\*DSM criteria

\*\*\*Hard drugs include cocaine, crack, Meth, opioid, stimulants, Rohypnol, GHB

final follow-up a significant minority of P18 participants continued to report past year stop-and-frisk, including one in nine White and non-Black participants and one in seven Black participants. The vast differences in the levels of self-reported stop-and-frisk experience in the P18 compared with per capita rates based on the NYPD records are consistent with documented underreporting in NYPD records [14], but also suggest that sexual minority men across race/ethnicity may experience much higher rates of police contact than non-sexual minority populations. We observed Black YSMM who lie at the intersection of racial minority and sexual minority status face the greatest vulnerability and that YSMM with the greatest economic and social vulnerability were most likely to report stop-and-frisk experience. Those with very high numbers of stop-and-frisk events demonstrated substantial psychosocial vulnerability highlighting how syndemics of stigma, internalized homophobia, and adverse mental health, in conjunction with criminal legal system contact, negatively affect some Black YSMM. This increased exposure was loosely associated with Black participants' residing in neighborhoods with histories of being heavily policed. Stop-and-frisk adds to the burden of structural and systematic discrimination that is known to degrade health, well-being and sense of safety in this already vulnerable group [26, 27]. Together, results indicating the pervasive experience of stop-and-frisk among YSMM, particularly Black YSMM, highlight the need for reduction of this stressful and deleterious life event among YSMM. The results underscore the

call for an alternative to policing, and a community response that addresses the economic, social, and heath care needs of the population.

While YSMM of all races and ethnicities face high rates of police contact, racial inequality persists within this vulnerable group. Black YSMM faced a significantly higher burden of repeat stop-and-frisk than their White counterparts. At baseline, nearly one-quarter of Black YSMM reported being stopped-and-frisked two or more times in the past year compared with one in five non-Black Hispanic/Latinx and approximately 10% of White participants. Further, the cumulative number of stop-and-frisk events in the past year per Black participants, was substantially higher than the stop-and-frisk rate among non-Black participants across follow-up. Race disparities appeared to increase over time as the experience became less normative overall. We also observed that the Black vs non-Black disparities were particularly pronounced in neighborhoods with overall lower levels of stop-and-frisk experience while in the highest stop neighborhoods non-Black Hispanic/Latinxs had a disproportionate rate of being stopped. Disproportionate rates of policing were observed among Black and Hispanic/Latinx partici-pants who did not endorse drug use, which is a primary reason motivating police stops.

The inequitable policing of racial/ethnic minorities observed in this study is consistent with the vast scientific literature and current media reports highlighting the inequitable policing including aggressive policing of Black individuals and communities [13, 28, 29]. Although the threat of police violence faced by Black Americans has recently gained prominence in public discourse, a long list of police-related deaths over the past 40 years highlights the long history of police related violence that has persisted into the modern era. Moreover, incidents which lead to death represent only a small fraction of violent police encounters; nonlethal physical force is far more common, if less consistently documented, and has been associated with a variety of adverse health and social outcomes [30].

We observed levels of stop-and-frisk based on self-report of the experience were dispropor-tionately high in this sample compared to the official NYPD stop-and-frisk counts. The sub-stantial magnitude of difference in reporting by participants compared to official records is consistent with concerns that official reporting by the NYPD does not fully reflect the burden of police practices on local populations. Moreover, our findings are suggestive of a heavy bur-den on YSMM, as has been suggested in qualitative studies and advocacy reports [31, 32]. There is agreement that levels of stop-and-frisk as a practice has become less common since the *Floyd v. City of New York* decision in 2013. However, the contrast between general popula-tion rates and the reports of P18 participants provide empirical evidence that the experience is likely far more common than official reports suggest.

In addition to a lack of information about the nature of the stop-and-frisk experience, which makes comparison to the NYPD official case counts of stop-and-frisk as well as uner-standing of the severity of the policing exposure challenging, additional limitations of the study include selection and measurement errors. First, attrition over cohort follow-up may have impacted measured levels of stop-and-frisk and race differences in stop-and-frisk levels if those who did not return for a study visit had different stop-and-frisk experience as those who presented for study visits. In addition, due to relatively modest sample size of the cohort, we collapsed multiple racial/ethnic categories and further omitted some (e.g., Asians) who did not identify as one of the three racial/ethnic groups most prominent in discussions of American policing. Although not explicitly limitations, we had no non-sexual minority group against which to compare levels of stop-and-frisk and additionally our analyses of correlates of stop-and-frisk were by design unadjusted to describe the social, mental health, and behavioral health needs of those who come in contact with the police. Our findings highlight the need for future studies in larger samples to include questions of stop-and-frisk experience to enable comparisons of YSMM to non-sexual minority youth. New York City does not report stop-

and-frisk rates by sexual minority status, making these comprisons somewhat limited. In addition, future studies should be powered to consider racial/ethnic differences in the burden of this exposure considering a heterogeneity of racial/ethnic groups compared YSMM and non-sexual minority individuals.

In conclusion, the current study highlights the substantial burden of stop-and-frisk police experience among YSMM in New York City, a group whose experience has remained under-studied and underdocumented, and a population that continues to face a heigtened burden of health disaprities including but not limted to HIV, mental helath burden, substance use, and violence [33, 34]. We observed that while rates of stop-and-frisk have declined over time, consistent with official NYPD reporting of the experience, levels remain high and are disproportionately high among Black YSMM. Findings that the most socially vulnerable YSMM are those most likely to experience stop-and-frisk policing highlight the need for community based solutions among those at risk of contact with the police as well as the need for institutions that do not explicitly focus on public safety (e.g., primary care physicians, schools) to raise awareness of police contact as a threat to health and wellbeing [35]. Doing so will reduce a key driver of adverse health in YSMM and particularly those who possess intersectional identities as both sexual minority and racial/ethnic minority individuals.

## Acknowledgments

We would like to acknowledge P18 participants for their participation in the Project 18 Cohort study and Natalia Irvine for her editorial contributions.

## Author Contributions

**Conceptualization:** Maria R. Khan, Farzana Kapadia, Amanda Geller, Kristen D. Krause, Charles M. Cleland, Typhanye V. Dyer, Danielle C. Ompad, Perry N. Halkitis.

**Data curation:** Farzana Kapadia, Amanda Geller, Medha Mazumdar, Kristen D. Krause, Richard J. Martino, Danielle C. Ompad.

**Formal analysis:** Medha Mazumdar, Danielle C. Ompad.

**Funding acquisition:** Maria R. Khan, Perry N. Halkitis.

**Investigation:** Farzana Kapadia, Perry N. Halkitis.

**Methodology:** Maria R. Khan, Farzana Kapadia, Amanda Geller, Joy D. Scheidell, Charles M. Cleland, Typhanye V. Dyer, Perry N. Halkitis.

**Project administration:** Farzana Kapadia, Perry N. Halkitis.

**Resources:** Maria R. Khan, Perry N. Halkitis.

**Supervision:** Maria R. Khan, Perry N. Halkitis.

**Validation:** Farzana Kapadia, Charles M. Cleland.

**Visualization:** Medha Mazumdar.

**Writing – original draft:** Maria R. Khan, Farzana Kapadia, Amanda Geller, Medha Mazumdar, Joy D. Scheidell, Charles M. Cleland, Typhanye V. Dyer, Danielle C. Ompad.

**Writing – review & editing:** Maria R. Khan, Farzana Kapadia, Amanda Geller, Medha Mazumdar, Joy D. Scheidell, Kristen D. Krause, Richard J. Martino, Charles M. Cleland, Typhanye V. Dyer, Danielle C. Ompad, Perry N. Halkitis.

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
