## [Decision Letter · Decision Letter 0]

28 Apr 2021

PONE-D-20-40991

Racial and ethnic disparities in "stop-and-frisk" experience among young sexual minority men in New York City

PLOS ONE

Dear Dr. Khan,

Thank you for submitting your manuscript to PLOS ONE. After careful consideration, we feel that it has merit but does not fully meet PLOS ONE’s publication criteria as it currently stands. Therefore, we invite you to submit a revised version of the manuscript that addresses the points raised during the review process.

Please address the comments of Reviewer number 2 and  Journal's requirements. 

We look forward to receiving your revised manuscript.

Kind regards,

Carmen Melatti, Staff Editor,

of behalf of 

Nickolas D. Zaller

Academic Editor

PLOS ONE

Journal Requirements:

2) Please state whether the data utilized in this study were de-identified/anonymised before access?

3) In statistical methods, please refer to any post-hoc corrections to correct for multiple comparisons during your statistical analyses. If these were not performed please justify the reasons. Please refer to our statistical reporting guidelines for assistance (https://journals.plos.org/plosone/s/submission-guidelines.#loc-statistical-reporting).

4) In your statistical analyses, please describe how you accounted for clustering of data by locality. For example, did you use or consider using multilevel models?

5)  We note that the grant information you provided in the ‘Funding Information’ and ‘Financial Disclosure’ sections do not match.

6)  We note that Figure 3 in your submission contain map images which may be copyrighted. All PLOS content is published under the Creative Commons Attribution License (CC BY 4.0), which means that the manuscript, images, and Supporting Information files will be freely available online, and any third party is permitted to access, download, copy, distribute, and use these materials in any way, even commercially, with proper attribution. For these reasons, we cannot publish previously copyrighted maps or satellite images created using proprietary data, such as Google software (Google Maps, Street View, and Earth). For more information, see our copyright guidelines: http://journals.plos.org/plosone/s/licenses-and-copyright.

5.1. You may seek permission from the original copyright holder of Figure 3 to publish the content specifically under the CC BY 4.0 license.

5.2. If you are unable to obtain permission from the original copyright holder to publish these figures under the CC BY 4.0 license or if the copyright holder’s requirements are incompatible with the CC BY 4.0 license, please either i) remove the figure or ii) supply a replacement figure that complies with the CC BY 4.0 license. Please check copyright information on all replacement figures and update the figure caption with source information. If applicable, please specify in the figure caption text when a figure is similar but not identical to the original image and is therefore for illustrative purposes only.

Reviewers' comments:

Reviewer's Responses to Questions

**Comments to the Author**

1. Is the manuscript technically sound, and do the data support the conclusions?

Reviewer #1: Yes

Reviewer #2: Yes

2. Has the statistical analysis been performed appropriately and rigorously? 

Reviewer #1: Yes

Reviewer #2: Yes

3. Have the authors made all data underlying the findings in their manuscript fully available?

Reviewer #1: Yes

Reviewer #2: Yes

4. Is the manuscript presented in an intelligible fashion and written in standard English?

Reviewer #1: Yes

Reviewer #2: Yes

5. Review Comments to the Author

Reviewer #1: Overall, I believe this is an important contribution to the literature on health and social disparities among sexual minority men. This is a well written paper and contains compelling findings for public health practice and public safety. The findings further document racial/ethnic inequities in policing and present new and compelling data related to community policing and sexual minority men.

Reviewer #2: You have provided data analysis for a space otherwise unresearched. Few, if any, authors are performing analysis and discussion on YSMM as it relates to stop and frisk. This manuscript has added an intersectional layer for race, gender, and other sociodemographic factors which makes the research distinct from extant knowledge. The results were congruent with findings from current literature regarding gender and race while complementing the area with data for YSMM. In the text, you explained an outlier that reported 66 stops in one year. More explanation on that person's experiences would be ideal in determining whether he/she/they had extenuating circumstances (such as activism, recidivism, etc) that may offer insight on other intersectional factors, which could possibly open the door for future research.

6. PLOS authors have the option to publish the peer review history of their article (what does this mean?). If published, this will include your full peer review and any attached files.

Reviewer #1: No

Reviewer #2: No

---

## [Author Response · Author response to Decision Letter 0]

12 Jul 2021

We have addressed all formatting inconsistencies.

2) Please state whether the data utilized in this study were de-identified/anonymised before access?

The data were de-identified (removal of participant name, address) with the exception of ZIP code of residence which enabled ZIP code-level analysis.

3) In statistical methods, please refer to any post-hoc corrections to correct for multiple comparisons during your statistical analyses. If these were not performed please justify the reasons. Please refer to our statistical reporting guidelines for assistance (https://journals.plos.org/plosone/s/submission-guidelines.#loc-statistical-reporting).

The paper was largely descriptive hence multiple comparisons were not considered. We aimed to describe the race disparities in stop-and-frisk among YSMM. We calculated RRs and 95% confidence intervals to describe differences between Black versus white and Latinx versus white participants on prevalence and rates of stop and frisk experience. We were not conducting multiple tests of significance.

4) In your statistical analyses, please describe how you accounted for clustering of data by locality. For example, did you use or consider using multilevel models?

Study participants were recruited from the New York City area. We accounted for geographic differences by mapping stop-and-frisk by geographic location and to compare race differences in levels of stop-and-frisk across different geographic settings defined by neighborhood-level stop-and-frisk rate. 

5) We note that the grant information you provided in the ‘Funding Information’ and ‘Financial Disclosure’ sections do not match.

We have addressed this inconsistency.

We have ensured the correct grant numbers have been indicated: National Institute on Drug Abuse grant ‘Stop-and-Frisk, Arrest, and Incarceration and STI/HIV Risk in Minority MSM’ (Principal Investigator: Maria Khan; R01 DA044037) and the P18 Cohort Study Grant (Principal Investigator: Perry Halkitis; 1R01DA025537 and 2R01DA025537). 

6) We note that Figure 3 in your submission contain map images which may be copyrighted. All PLOS content is published under the Creative Commons Attribution License (CC BY 4.0), which means that the manuscript, images, and Supporting Information files will be freely available online, and any third party is permitted to access, download, copy, distribute, and use these materials in any way, even commercially, with proper attribution. For these reasons, we cannot publish previously copyrighted maps or satellite images created using proprietary data, such as Google software (Google Maps, Street View, and Earth). For more information, see our copyright guidelines: http://journals.plos.org/plosone/s/licenses-and-copyright.

Figure 3 was created by the authors using P18 participant data. It is not a copyrighted image.

Reviewer #1: Overall, I believe this is an important contribution to the literature on health and social disparities among sexual minority men. This is a well written paper and contains compelling findings for public health practice and public safety. The findings further document racial/ethnic inequities in policing and present new and compelling data related to community policing and sexual minority men.

We appreciate the enthusiasm for this work.

Reviewer #2: You have provided data analysis for a space otherwise unresearched. Few, if any, authors are performing analysis and discussion on YSMM as it relates to stop and frisk. This manuscript has added an intersectional layer for race, gender, and other sociodemographic factors which makes the research distinct from extant knowledge. The results were congruent with findings from current literature regarding gender and race while complementing the area with data for YSMM. In the text, you explained an outlier that reported 66 stops in one year. More explanation on that person's experiences would be ideal in determining whether he/she/they had extenuating circumstances (such as activism, recidivism, etc) that may offer insight on other intersectional factors, which could possibly open the door for future research.

We have had internal discussion about how to best address the comment given some of our growing concerns about deductive disclosure of this individual as a result of providing too much information on him. Since the reviewer expressed interest in intersectional stigma we elected to provide some details on the individuals' psychosocial vulnerability and PTSD risk but avoided additional demographic, socioeconomic factors, sex trade involvement etc. We added this information to the results and a note in the discussion to reflect on results.

---

## [Decision Letter · Decision Letter 1]

3 Aug 2021

Racial and ethnic disparities in "stop-and-frisk" experience among young sexual minority men in New York City

PONE-D-20-40991R1

Dear Dr. Khan,

We’re pleased to inform you that your manuscript has been judged scientifically suitable for publication and will be formally accepted for publication once it meets all outstanding technical requirements.

Kind regards,

Nickolas D. Zaller

Academic Editor

PLOS ONE

Additional Editor Comments (optional):

Reviewers' comments:

Reviewer's Responses to Questions

**Comments to the Author**

1. If the authors have adequately addressed your comments raised in a previous round of review and you feel that this manuscript is now acceptable for publication, you may indicate that here to bypass the “Comments to the Author” section, enter your conflict of interest statement in the “Confidential to Editor” section, and submit your "Accept" recommendation.

Reviewer #1: All comments have been addressed

2. Is the manuscript technically sound, and do the data support the conclusions?

Reviewer #1: Yes

3. Has the statistical analysis been performed appropriately and rigorously? 

Reviewer #1: Yes

4. Have the authors made all data underlying the findings in their manuscript fully available?

Reviewer #1: Yes

5. Is the manuscript presented in an intelligible fashion and written in standard English?

Reviewer #1: Yes

6. Review Comments to the Author

Reviewer #1: Thank you for your responsiveness to the reviewer comments. All the comments have been adequately addressed.

7. PLOS authors have the option to publish the peer review history of their article (what does this mean?). If published, this will include your full peer review and any attached files.

Reviewer #1: **Yes: **Jordan J White

---

## [Editor Report · Acceptance letter]

18 Aug 2021

PONE-D-20-40991R1 

Racial and ethnic disparities in “stop-and-frisk” experience among young sexual minority men in New York City 

Dear Dr. Khan:

I'm pleased to inform you that your manuscript has been deemed suitable for publication in PLOS ONE. Congratulations! Your manuscript is now with our production department. 

Kind regards, 

on behalf of

Dr. Nickolas D. Zaller 

Academic Editor

PLOS ONE